# Characteristics and Antioxidant Potential of Cold-Pressed Oils—Possible Strategies to Improve Oil Stability

**DOI:** 10.3390/foods9111630

**Published:** 2020-11-08

**Authors:** Magdalena Grajzer, Karolina Szmalcel, Łukasz Kuźmiński, Mateusz Witkowski, Anna Kulma, Anna Prescha

**Affiliations:** 1Department of Food Science and Dietetics, Wroclaw Medical University, Borowska 211, 50-556 Wroclaw, Poland; mateusz.witkowski@umed.wroc.pl (M.W.); anna.prescha@umed.wroc.pl (A.P.); 2Student Scientific Club at Food Science and Dietetics Department, Wroclaw Medical University, Borowska 211, 50-556 Wroclaw, Poland; karolina.szmalcel@student.umed.wroc.pl; 3Department of Process Management, Management Department, Wroclaw University of Economics, 50-556 Wroclaw, Poland; lukasz.kuzminski@ue.wroc.pl; 4Department of Genetic Biochemistry, University of Wroclaw, Przybyszewskiego 63/77, 51-148 Wrocław, Poland; anna.kulma@uwr.edu.pl

**Keywords:** antioxidants, antioxidant capacity, cold-pressed seed oils, oxidative stability, shelf life, lipid oxidation

## Abstract

The relations of the antiradical capacity to oxidative stability parameters and the contents of fatty acids, sterols, tocopherols, phenols, flavonoids, chlorophyll, Cu, and Fe were assessed in 33 cold-pressed seed oils: Walnut (7 brands of oils), rosehip (3), camelina (6), milk thistle (5), flax (6), and pumpkin (6). The antiradical capacity of oils depended strongly on tocopherol contents with a synergistic effect with polyphenols. The efficacy of tocopherols in cold-pressed oils was accompanied by a negative correlation of their antioxidant capacity with the peroxide value increase after 3 months of shelf life. This study also showed a positive correlation between the content of phytosterols and the antiradical capacity in the lipophilic fraction of cold-pressed oils rich in n-3 polyunsaturated fatty acids (PUFAs). Multiple regression analysis identified groups of antioxidants naturally occurring in cold-pressed oils in relation to their fatty acid composition, which added to the cold-pressed oils could provide possible strategies to improve their stability. Achieving high stability is primarily a result of high phytosterol content exceeding the molar ratio of 1:100 for total phytosterols to α-linolenic acid. However, the molar ratios of tocopherols to linoleic acid below 1:2000 and polyphenols to linoleic acid below 1:3000 does not prevent oxidation in oils with the predominance of linoleic acid.

## 1. Introduction

Cold-pressed oils are continuously gaining in popularity among consumers. The advantage of cold-pressed oils over refined ones is largely due to the numbers of bioactive substances recovered in the pressing process. These include antioxidants such as tocopherols, polyphenols, and squalene, which have been proven to retard lipid oxidation [1,2]. For cold-pressed oils rich in polyunsaturated fatty acids (PUFAs), the effectiveness of antioxidants is of particular importance, as high susceptibility of PUFAs to oxidation entails a risk of oil deterioration and, consequently, detrimental health effects when it is consumed [3]. It has been shown that oil oxidation mechanisms and rates can vary not only with the fatty acid profile but are also influenced by antioxidant polarity, structure, concentration, and mutual ratios [4,5]. The contribution of individual antioxidants to the oxidative stability of cold-pressed oil depends on the PUFA ratio in oil as well as the pro/antioxidant composition [1]. Owing to the low degree of processing, cold-pressed oils contain more components derived from the raw material that have a pro-oxidant effect in oil, among them metal ions, chlorophylls, and lipid peroxides [4]. In contrast to refined oils, cold-pressed oils often have a higher initial autooxidation state and less predictable oxidative stability. The pro- and anti-oxidant interactions may be reflected in the oil’s antioxidant activity, which would affect its shelf-life. A number of methods have been developed for the assessment of the antioxidant capacity of oils [6]. Among them, the ability of antioxidants in oil to reduce stable 2,2-diphenyl-1-picrylhydrazyl (DPPH^∙^) radical (DPPH assay) was adopted for the determination of the antiradical activity of hydrophilic and lipophilic antioxidants in oils [5,6,7,8]. A strong relation between the content of phenolic compounds in the oil and DPPH was found for the hydrophilic fraction of olive oil and several seed oils [7,9]. Correlations of DPPH values in non-fractionated maize and soybean oil with tocopherol contents were reported, indicating the major role of these lipophilic antioxidants in antiradical activity of oils [6,7].

On the food market, there are cold-pressed oils with different proportions of PUFAs and various compositions of antioxidants. These oils, however, are prone to autoxidation, resulting in the accumulation of considered unhealthy products of lipid oxidative deterioration. Thus far, there have been proposed many strategies to ease the progress of lipid oxidation in edible oils [1,10,11,12,13]. These include, among others, fortification with antioxidants, modifying oil manufacturing technology, and a combination of highly unsaturated oils with a less unsaturated fatty acid oil. However, the effectiveness in maintaining the oxidative stability of naturally occurring compounds in oils is rarely investigated. The influence of cold-pressed oil composition and antioxidative potential on its oxidative stability during shelf life has also thus far been poorly studied.

For this study, we have chosen high PUFA (>50%) oils representing a wide range of n-6/n-3 ratio (from <1 to >50), which were produced by cold pressing: Walnut, rosehip, camelina, milk thistle, flaxseed, and pumpkin oil. Each of these oils has a characteristic antioxidant profile, including the composition and amount of tocopherols and sterols, and they also contain different amounts of polyphenolic compounds. These oils are already greatly appreciated by consumers for their nutritive and organoleptic values. The beneficial health effects of these seed oils have been reported. Camelina, milk thistle and rosehip oils improved the serum lipid profile. Moreover, milk thistle oil reduced the paracetamol hepatotoxicity, and rosehip oil had positive effects on various skin diseases [14,15,16]. Walnut and flaxseed oils have been shown to exert cardioprotective effects [17,18]. Pumpkin seed oil prevented and inhibited the development of benign prostate hyperplasia [19].

The identification of the associations that influence antioxidant potential may enable us to predict the specificity of oxidative changes and discover potential antioxidant mixtures, which could prolong antioxidant stability in cold-pressed oils. Improved oxidative stability of cold-pressed oils through adding molecules naturally occurring in oils could result in a product that is lower in potentially harmful oxidized lipids. For this reason, the object of this study was first to determine the composition and antiradical activity of commercially available cold-pressed walnut, rosehip, camelina, milk thistle, flaxseed, and pumpkin oils and their oxidative stability parameters during shelf life. Secondly, to analyze the interrelations of the results obtained and, finally, to recognize the most effective antioxidant compositions for an individual oil’s PUFA profile.

## 2. Materials and Methods

### 2.1. Oil Samples

Thirty-three cold-pressed oils freshly produced from walnut (*Juglans regia* L., *n* = 7), and seeds of rosehip (*Rosa canina* L., *n* = 3), camelina (*Camelina sativa* L., *n* = 6), milk thistle (*Silybum marianum* L., *n* = 5), flax (*Linum usitatissimum* L., *n* = 6), and pumpkin (*Cucurbita pepo* L., *n* = 6) were obtained from 14 manufacturers in Poland. Oils provided in glass bottles were analyzed shortly after delivery (spectrophotometric and titration methods). Samples of oils were stored at −20 °C for chromatographic analyses. The oxidative stability parameter analyses were also performed in oils after 3 months of shelf-life storage at room temperature at 22 °C ± 1 °C. During the shelf-life storage, the oils were kept in the dark or exposed to cold fluorescent light in a cycle of 12 h light, alternating with 12 h darkness. All of the oil samples were prepared and determined in triplicate to achieve a coefficient factor under 5%.

### 2.2. Fatty Acid Composition

Fatty acid methyl esters (FAMEs) were prepared in accordance with the method developed by Prescha, et al. [20]. The gas chromatographic separation of FAMEs was performed on a SPTM-2560 fused silica capillary column (Supelco, USA, 100 m × 0.25 mm × 0.2 μm) using an Agilent GC 6890N gas chromatograph coupled with a FID detector (Agilent Technologies, Santa Clara, CA, USA). Detailed chromatographic separation conditions and the program were described previously [21].

### 2.3. Phytosterol and Squalene Content

Silyl derivatives of sterols and squalene were prepared in accordance with the method described by Shukla et al. [22]. The gas chromatographic separation of compounds was performed on an Elite-17ms capillary column (PerkinElmer, Waltham, MA, USA, 30 m × 0.25 mm × 0.25 μm) using an Agilent GC 7890B gas chromatography coupled with a 7000D mass spectrometer (Agilent Technologies, Santa Clara, CA, USA). Helium was used as a carrier gas at a flow rate of 1.1 mL/min, and the separation was carried out at a temperature set from 120 °C (for 1 min) to 290 °C (5 for min); the temperature increased at a rate of 5 °C/min. The source and transfer line temperatures were 230 °C and 290 °C, respectively. The quantifications were made on a 6890 N gas chromatograph (Agilent Technologies, Santa Clara, CA, USA) equipped with an FID detector and an Elite-17ms capillary column (PerkinElmer, Waltham, MA, USA, 30 m × 0.25 mm × 0.25 μm). 5α-cholestane was used as an internal standard for quantitative analysis, and Chemstation v. B.04.02 was used to calculate the results. In the case of coeluted peaks Amdis ver. 2.66 (NIST, Gaithersburg, MD, USA) was used with a specific base of mass spectra created from pure standards (.ims and .isl) to separate mass spectra as well as to identify and quantify the co-eluted molecules.

### 2.4. Tocopherol Content

The tocopherol (α, γ, δ) content was determined after oil samples were saponified according to a protocol described previously [23] and then separated on an Acquity CSH C18 column (1.7 µm, 100 × 1.0 mm, Waters, Milford, MA, USA) using the Acquity Waters UPLC system with PDA (Waters, Milford, MA, USA) [24].

### 2.5. Total Phenol and Flavonoid Content

To determine the total phenol and flavonoid content, oil samples (2.5 g) dissolved in 5 mL of hexane were extracted successively with three portions of 90% aqueous methanol (3.5 mL). The hydrophilic layer, filtered through the 0.45 μm hydrophilic polytetrafluoroethylene (PTFE) syringe membrane filter (Merck Millipore Poland), was collected in a round flask and brought to dryness in a vacuum rotary evaporator at 38 °C. The residue was dissolved in 1.5 mL of methanol and stored at −20 °C prior to analysis. The spectrophotometric determination of total phenol content was carried out according to Siger et al. [25], and total flavonoids were measured according to the method of Choo et al. [26].

### 2.6. Determination of Chlorophyll Content

The total chlorophyll content was determined spectrophotometrically. The oil sample without any dilution was placed in a quartz cuvette, and the absorbance was measured at 670 nm. The result was corrected for the background absorption, and the content was calculated in reference to the standard calibration curve of pheophytin a, which is the main pigment in crude vegetable oils [7].

### 2.7. Determination of Cu and Fe Content

Cu and Fe contents were determined using electro-thermal atomic absorption spectrometry. An oil sample (0.5 g) was pre-treated with 4 mL mixtures of 65% HNO_3_ (*v*/*v*) and left for 24 h. Then 1 mL of 30% H_2_O_2_ (*v*/*v*) was added to the sample, and it was mineralized in a microwave-assisted digestion oven (Millestone SRL, Sorisole BG, Italy). The total mineralization of samples was achieved in 2 h and 30 min. The mineralized sample was diluted up to 10 mL with deionized water (18 MΩ) and analyzed using a Perkin Elmer atomic absorption spectrometer (PinAAcle 900T, Waltham, MA, USA) equipped with a trans heated graphite furnace. The procedure has already been applied and described thoroughly previously [24].

### 2.8. Antiradical Scavenging Activity

The antiradical activity of oils and their hydrophilic and lipophilic fractions was established by the 2,2-diphenyl-1-picrylhydrazine (DPPH) assay according to the method already described and developed by Prescha et al. [27]. To obtain the hydrophilic and lipophilic fraction, 500 μL of oil was mixed with 500 μL of methanol and then centrifuged to allow the fraction to separate. The methanol fraction was collected into an Eppendorf tube up to a volume of 500 μL. The rest of the sample consisted of the lipophilic fraction.

### 2.9. Oxidative Stability Parameters of Oils

Acidic value (AV), peroxide value (PV), *p*-anisidine value (*p*-AV), and conjugated trienes were determined using CEN ISO Polish official methods. Spectrophotometric determination of conjugated dienes was performed according to a method already described [7].

### 2.10. Statistical Analysis

Data for each kind of oil were reported as a median of each manufacturer’s oil examined. Statistical analysis of the relation between antiradical activity values and oil components was primarily performed in correlation charts to select explanatory variables (correlation coefficient analysis method). Critical values of the correlation coefficient were: α = 0.05, t_31; 0.05_ = 2.04, r* = 0.34. For further statistical analysis, multiple regression was performed. Statistical significance was declared at *p* < 0.05.

## 3. Results

### 3.1. Fatty Acid and Phytosterol Composition

The composition of fatty acids in studied oil is shown in Table 1. Rosehip oil was characterized by the highest content of PUFAs (ca. 75%) and was one of the richest in α-linolenic acid (ALA) together with linseed and camelina oils. Analyzed rosehip oils also contained the lowest level of saturated fatty acids (SFAs) (7.1%), whereas camelina oils comprised the smallest overall median quantity of monounsaturated fatty acids (MUFAs). Pumpkin and milk thistle oils contained the lowest amount of PUFAs (less than 60%), with a predominance of linoleic acid (LA). Among investigated oils, the highest median total percentages of both SFAs and MUFAs were detected in pumpkin oils and amounted to 18.2% and 30.8%, respectively. Milk thistle oils were characterized by the most diverse composition of acids. High ALA content resulted in a very low n-6/n-3 ratio calculated for camelina and linseed oils, while very high values of n6/n3 ratio reflected the predomination of LA and negligible content of ALA in milk thistle and pumpkin oils.

The phytosterol content in oils is shown in Table 2. The highest median values of total phytosterols, exceeding 5 g/kg, were determined in rosehip and flaxseed oils. In most oils, with the exception of pumpkin oil, β-sitosterol was the predominant phytosterol, the greatest amount of which was found in rosehip oils (4314.2 mg/kg). Pumpkin oils contained mainly Δ7-sterols (up to 98%), and considerable amounts of Δ7-sterols were also found in milk thistle oils (37%).

### 3.2. Antioxidants, Other Minor Components and Antiradical Scavenging Activity of Oils

Table 3 shows the amounts of minor compounds known to affect antioxidant potential and oxidative stability of studied oils. Among antioxidants, three forms of tocopherol were detected, of which γ-tocopherol predominated in almost all studied oils except milk thistle oils containing mainly the α form. Rosehip oils, which contained the highest total amount of tocopherols (1036 mg/kg), were also distinguished by a considerable amount of δ-tocopherol (253 mg/kg). The least abundant in tocopherols was milk thistle oil.

Among other antioxidants investigated in the study, the highest amounts of both flavonoids and squalene were found in pumpkin oils. Although the highest median total phenol level was determined in camelina oils (117.3 mg CAE/kg), the differences between samples from different brands were notable.

Transition metals such as iron or copper and chlorophyll may negatively affect the oxidative stability. Metal increases the rate of oil oxidation due to the reduction of activation energy of lipid autoxidation. Chlorophylls and their degradation products also accelerated the lipid oxidation in the presence of light and atmospheric oxygen [4]. Therefore, the amounts of components that may negatively affect oxidative stability were also measured. Pumpkin oils contained the most chlorophylls (6.63 mg/kg) and iron (0.4 mg/kg) among the studied oils. The highest amount of Cu was noted in milk thistle oil (0.037 mg/kg). In other oils, the median level of Cu did not exceed 0.015 mg/kg.

Figure 1 shows the results of the antiradical scavenging activity assay in oils as well as in both hydrophilic and lipophilic fraction of oils. The DPPH test showed the highest antioxidant potential in the non-fractionated rosehip oils and then camelina oils of 2.95 and 2.40 mM Trolox equivalent antioxidant capacity (TEAC)/kg, respectively. The results for the lipophilic fraction of oils were similarly distributed (1.87 and 1.60 mM TEAC/kg, respectively). The antiradical potential of the hydrophilic fraction of oils was much lower than that of the lipophilic fraction, with the highest median value obtained for rosehip oils (0.34 mM TEAC/kg).

### 3.3. Oxidative Stability Parameters of Fresh and Stored Oils and Correlation with Studied Oil Characteristics

The parameters of quality and oxidative stability of oils at the start point and after 3 months of shelf life are presented in Figure 2. The acid value of oils of up to 2 mg KOH/kg (walnut oil) had not changed after 3 months, or increased only slightly. Most of the hydroperoxides in fresh oil were found in rosehip oil (PV = 0.76 mEq O_2_/kg), but this was not associated with the aggravation of their accumulation during the shelf-life test. The largest increase in peroxide value was observed in camelina and walnut oils as well as in one sample of flaxseed oil. Among fresh oil samples, the highest *p*-anisidine values were observed for walnut and rosehip oils (median *p*-AV about 1.7), of which the former accumulated the most secondary oxidation products after 3 months. The conjugated products of linoleic and linolenic acids were also determined. At the start point of the shelf life test the highest conjugated diene and triene concentrations were noted in pumpkin oil (5.03 and 1.81 µmol/g, respectively).

The involvement of oils’ characteristics in their antioxidant capacity and oxidation stability was statistically analyzed in order to establish the main influence determining the oxidation changes of cold-pressed oils (Table 4 and Table 5. The antiradical activity of oils measured in the DPPH test was shown to be strongly dependent on the total tocopherol contents, and a similar apparent relation was found for polyphenols in the case of the DPPH values of the hydrophilic fraction (Table 4). A positive correlation was found between the content of phytosterols, and the values of DPPH determined for the lipophilic fraction of oils. The increase of peroxide value after 3 months of shelf life of oils was inversely linked to the value of the DPPH in non-fractioned oils (Table 4).

We also established the relationship between the amounts of antioxidant and other minor components affecting oil’s stability and DPPH antioxidant potential of non-fractionated oils with regard to oil kinds (and thus oils’ fatty acid characteristics) (Table 5). The statistical analysis revealed that tocopherols had a significant impact on the antioxidant potential in rosehip, camelina, milk thistle, and pumpkin oils. Antioxidant potential in walnut oils was impacted by polyphenols and flavonoids. Flaxseed oils antioxidant potential depended only on phytosterol content.

## 4. Discussion

### 4.1. Fatty Acid and Phytosterol Composition

It is well known that the tendency of oils to undergo oxidation increases with the number of double bonds present in fatty acid molecules [4]. Obtained high ALA content in studied rosehip and linseed oils agrees with previous studies [24,27], while amounts of ALA determined in camelina oil exceed previously reported values [29]. Quantities of PUFAs in pumpkin and milk thistle oil are in accordance with previously published data [27,30]. In the cited works, a higher n-6/n-3 ratio was found for these oils: Up to 440 for milk thistle and up to 117 for pumpkin oils, which resulted mainly from the lower level of ALA detected in the samples. According to recommendations for essential fatty acid intakes developed for adults and children, walnut oils were shown to have a beneficial LA/ALA ratio: ca. 5 [31]. However, an increasing body of data indicates that the total intake of n-3 fatty acids was more important than the n-6/n-3 ratio [32]. Consuming oils abundant in ALA—flaxseed, camelina, and rosehip oil—may considerably enrich a modern diet deficient in n-3 fatty acids. Therefore, the stability of these oils must be a matter of concern due to the susceptibility of ALA to oxidation.

The amounts of total phytosterol in the studied oils differed by up to 4.3 times, with pumpkin oil containing the most of these compounds. The results of phytosterol analysis are in accordance with data reported for these oils by other authors [24,33]. The structure and concentration-dependent contribution of phytosterols to inhibiting the oxidation of PUFA have previously been demonstrated [34,35]. At high concentrations, their efficacy may be comparable to proven lipophilic antioxidants, although the mechanism of action has not been thoroughly examined. The addition of campesterol and β-sitosterol (at a concentration of 6.7 or 13.5 µM in acetonitrile) individually and in the natural phytosterol mixture (β-sitosterol and campesterol ratio 3.3:1) resulted in significant inhibition of methyl linoleate oxidation (1 mM) [36]. The authors of the mentioned study did not investigate the effect of sterols on ALA oxidation; however, it can be noted that there was a similar ratio of the molar sum of sterols to PUFA in walnut oils analyzed in our study, as the highest one applied in the cited work. In the other oils, we analyzed the amounts of sterols per 1 mM PUFA were much higher, from ca. 0.03 mM in camelina oil to 0.07 mM in pumpkin seed oil, respectively. 

### 4.2. Antioxidants, Minor Components, and Antiradical Scavenging Activity of Oils

Tocopherols have proven to be important contributors to the antioxidant activity of oils rich in PUFA [1]. The capacity of tocopherols in scavenging radicals and metal chelating ability enables high efficiency to be achieved in oils. However, prooxidant effects may occur in high concentrations of α-tocopherol [37]. The optimal level of alpha-tocopherol in bulk oil has been shown to be 100 mg/kg, while the best antioxidant effect of gamma-tocopherol is achieved at a concentration of 250–500 mg/kg bulk oil, and no pro-oxidant effect of this homolog can be shown in a wide range of content in oils [10]. Tocopherol at a concentration of 50 mM has been shown to have twice the free radical scavenging activity of 10 mM. Moreover, during shelf-life storage, the decrease of its concentration is much higher in oils than gamma tocopherol, which strongly correlates with the formation of lipid hydroperoxides [38]. The α-tocopherol content in our studied oils was low, from undetectable (camelina oil) to 130 mg/kg, except for milk thistle, where this homolog prevailed and its amount was more than twice the one considered optimal. The remaining oils were dominated by γ-tocopherol in quantities from about 200 to 670 mg/kg. Oils rich in tocopherols, i.e., rosehip and camelina oils, also contained δ-tocopherol, which in some studies showed stronger antiradical properties than gamma homologs [12]. The analyses of tocopherols have produced similar results to those previously published [24,30].

Phenolic compounds may act as reducing agents, singlet oxygen quenchers, hydrogen donors, and metal chelators, which translates into their potency as antioxidants. In many studies, phenolics were shown to be responsible for the stability of high PUFA oil during storage [1,37,39]. Of the phenolic compounds in oils, phenolic acids are mainly present, and among them, hydroxycinnamic acid derivatives have higher antioxidant activity in lipid systems than hydroxybenzoates [1]. Numerous studies have shown that phenolic acids can be effective antioxidants in oils individually or in different proportions and also in mixtures with tocopherols despite their poor solubility in the oily phase. Their effectiveness depends on the concentration in oils. It was observed that caffeic and sinapic acid added to methyl linoleate in a concentration of 1000 µM cause about 10 times lower formation of hydroperoxides than in a concentration of 50 µM. In higher concentrations, its activity exceeds that obtained for α-tocopherol [11]. An even higher antioxidant activity in bulk methyl linoleate than caffeic acid and α-tocopherol is shown by protocatechuic aldehyde at a concentration of 0.015–0.08 mg/kg, with its scavenging kinetic activity being much slower than that of caffeic acid [13]. At concentrations above 1000 µM (ca. 180 mg/kg), 3-hydroxybenzoate may have pro-oxidative effects on PUFAs [11]. The phenolic content of the studied oils was within the range of values for which the positive effect of phenolic acids on PUFA was revealed (Table 3). Camelina and pumpkin oils were characterized by the largest median phenolic amounts, although the differences between the samples of oils from particular producers were considerable. In oil from camelina seeds grown in Slovenia [29], as much as 128 mg/kg of total phenolic content expressed as chlorogenic acid equivalents were reported, but the amounts were not directly comparable due to the different compounds used for the calibration curve. The content of phenolics measured in pumpkin oil was not in accordance with any published data. Studies by Andjelkovic et al. [40] and Siger, Nogala-Kalucka, and Lampart-Szczapa [25] showed a two to four times higher phenolics level of pumpkin seed oil, in contrast to a study by Parry, Hao, Luther, Su, Zhou, and Yu [9] reporting 10 times lower levels of these compounds. Walnut oil also showed a relatively excessive amount of phenolics. In a recent study, lower amounts of polyphenols than in the present work, not exceeding 65 mg GAE/kg, were found in cold-pressed walnut oils from China [41], while authors from Turkey noted about 10 times lower amounts of these compounds [42]. Published data regarding total phenolics in milk thistle oil are scarce. Parry, Hao, Luther, Su, Zhou, and Yu [9] reported 40 times higher levels of these compounds in samples of milk thistle oils produced in the USA. In flaxseed oil, the content of phenolics was found to be the lowest, which was nevertheless several times higher than the figures reported by Siger, Nogala-Kalucka, and Lampart-Szczapa [25].

The antioxidant activity of flavonoids depends on the pattern of hydroxyl substituents in their molecules, which are able to donate hydrogens to peroxyl radicals. The high effectiveness of selected intermediate polarity plant flavonoids, such as quercetin, in preventing oxidation of n-3 and n-6 PUFA rich oils has been observed. In concentrations from 50 to 400 µM (15–120 mg/L), quercetin exerts similar effectiveness in free radical scavenging, but its chelating properties are highest in a concentration of 100 µM [37]. Total flavonoid content in tested cold-pressed oils taken for testing was low and did not exceed 17 mg LE/kg, except for pumpkin seed oil, which was shown to be relatively rich in these compounds (Table 3). The data regarding the flavonoid content in cold-pressed oils are very limited. Macedonian flaxseed oil has been shown to contain about three times more flavonoids than samples analyzed in this study [43]. Interestingly, for flaxseed oils from New Zealand, much higher flavonoid levels values (up to 256 mg LE/kg) were noted than values obtained in this study [26,44].

Among substances exerting antioxidant activity, squalene content was measured in oils. Squalene acts as a scavenger of peroxyl radicals and has been shown to inhibit oxidation of both n-3 and n-6 fatty acids with high efficacy [45]. Squalene was prominent in pumpkin oils, and quite a high amount was also found in rosehip oils (Table 3). Although the results of the squalene determinations are generally in line with the literature, there are reports on squalene content in pumpkin oil up to 6 times higher than found in this study [7,46]. To obtain a high squalene content in pumpkin oil, the precise manufacturing conditions play a crucial role. Generally, husking and appropriately heating up seeds just before pressing results in oils with a higher squalene level [47].

Chlorophylls in oil can act as antioxidants in the absence of light, and they are also strong photosensitizers in the oxidation process [4]. Therefore, their effect on the shelf life of tested oils can be bidirectional, depending on whether the oil was packed in dark or light glass. In the studied pumpkin oils, considerably more chlorophylls were found than in other samples (Table 3). However, in the samples of most of the examined oils from various producers, quite a wide range of obtained values was found. As can be seen from the comparison with other publications, the content of these compounds in oils may vary, which can be attributed to both the quality characteristics of seeds and the purification processes [4,7].

Transition metals-iron or copper-present in cold-pressed oils lead to lipid alkyl radical production and increase the rate of oil oxidation [4,48]. Literature data usually report higher concentrations of Fe ions in oils compared to Cu, as confirmed by the results obtained in this study (Table 3). According to reports of other authors, the contents of these elements may vary considerably, which is attributable to the degree of processing (filtering, decanting) to which crude oil is subjected [1,4].

The presence of antioxidants and prooxidants translates into the antiradical activity of oils. Detected in non-fractioned rosehip and camelina oils, the highest antioxidant potential and distribution of lipophilic fraction results clearly corresponds to the prominent tocopherol content in these oils (Figure 2). The significantly lower potential of the hydrophilic fraction is a result of the poorer representation of hydrophilic antioxidants in oils. It should be noted that previously published values obtained in the same DPPH protocol were usually lower, both for oils tested in this paper and for their non-polar fraction [5,7].

### 4.3. Oxidative Stability Parameters of Fresh and Stored Oils and Correlation with Studied Oil Characteristics

All examined oils were well within the permissible acid and peroxide value limits quoted for cold-pressed and virgin oils under the Codex Alimentarius Commission standards [49]. The high conjugated diene and triene concentration noted in pumpkin oil at the start of the shelf life may result from the co-occurrence of some compounds absorbing in the same wavelength range region, such as carotenoids present in pumpkin seeds [50]. In the other tested oils, the values recorded for conjugated products of oxidation were low, and only in flaxseed oil was a marked increase in triene concentration found.

Statistically assessed dependence of antiradical activity (DPPH test) on tocopherol content and DPPH hydrophilic values on polyphenol concentration explicitly indicates the dominant contribution of tocopherols to the antiradical efficiency of oils, whose effectiveness in neutralizing free radicals in oil has been proven [8,50]. An analogy between the antiradical activity of the hydrophilic fraction and total phenolic content was also found in previous studies [46,50]. The statistical analysis indicated that tocopherol activity is considerably supported by the presence of polyphenols in the oil. Synergism of polyphenols as metal chelators and tocopherols acting as radical scavengers has been demonstrated in previous studies for high PUFA oils [51]. The potency of polyphenols can be seen in both ALA- and LA-rich oils, as shown by the results of the multiple regression analysis presented in Table 5, where such relations can be observed for walnut, camelina, and milk thistle oils. The oils examined were found to contain low amounts of peroxides, thus the beneficial effect of tocopherol-polyphenol synergism was not affected by these primary oxidation products [1,8]. The relevance of the interaction between polyphenols and tocopherols in antioxidant capacity also confirms the lack of correlation between tocopherols and DPPH values of the lipophilic fraction deprived of polyphenols (Table 5). The efficacy of tocopherols acting in non-fractionated cold-pressed oils was accompanied by a negative correlation of their antioxidant capacity with the increment of peroxide value after 3 months of shelf life (Table 4). Interestingly, a positive correlation was also found between the antioxidant capacity of the hydrophilic fraction and tocopherols, which is probably due to the fact that oils containing a high proportion of tocopherols were also rich in polyphenols. For individual tocopherol homologs the concentrations effective for antioxidant activity vary; therefore, the beneficial effect of the tocopherol-polyphenol ratio was also observed in milk thistle oil, which contains α-tocopherol showing antioxidant activity only at low concentrations [1,2]. Targeting of tocopherol homologs on individual PUFAs was evident in the present study. This is in accordance with multivariate analysis of 14 vegetable oils which showed natural interrelations between α-tocopherol and LA, on the one hand, and between γ-tocopherol and ALA, on the other hand [1,2].

This study also showed a positive correlation between the content of phytosterols and the values of DPPH in the lipophilic phase of cold-pressed oils (Table 5). The ability of selected sterols to inhibit oxidation have been previously revealed in lipid matrix, including bulk oils [8,52]. As can be seen in Table 5, a significant effect of sterols on the antioxidant potential was observed for oils containing a considerable amount of these phytosterols, which show a significant inhibitory effect on oxidation propagation: β-sitosterol in rose hip oil and also Δ5-campesterol in flaxseed oil. In both oils Δ5-avenasterol was also present unless the rest of the studied oils and its ability to scavenge free radicals contributed to the relation to DPPH values [34]. It is worth underlining that flaxseed oil after 3 months showed negligible growth in oxidative stability parameters (the most in conjugated triene formation, however, their absolute content was still very low). Multiple regression analysis also revealed that polyphenols and flavonoids had a significant contribution to the antioxidant potential in walnut oil, but no influence of tocopherols, of which there were relatively few in this oil, was noted. This resulted in a smaller effect inhibiting the oxidation propagation in this oil and a noticeable increase in the formation of primary and secondary oxidation products after 3 months of shelf life.

With the rise of the unsaturation of the studied oils and increase of the linolenic acid content, other antioxidants seemed to share a role in antioxidant potential, either lipophilic (rosehip oil) or hydrophilic (walnut and camelina oil). In the camelina oil, the antioxidant potential seemed to be influenced by three groups of antioxidants: tocopherols, polyphenols, and flavonoids. The presence of other antioxidants—phenolic compounds, flavonoids—may synergize with tocopherols and minimize their loss of efficacy [53,54].

To summarize our findings, we consider three of the outcomes from our studies the most important observations, which can add novel information to the field of oils’ oxidative stability.

First of all, the large amount of phytosterols (ca. 5 g/kg) and possibly the presence of potent antiradical phytosterols such as Δ5—avenasterol (300–400 mg/kg) as well determine oxidative resistance in oils rich in PUFA (70%), where at least half of their PUFA composition consists of ALA. It seems that a molar ratio of 1:100 for total phytosterols to ALA and 1:1000 for Δ5—avenasterol to ALA is favorable for the storage stability of cold-pressed oil rich in this fatty acid. In our study, flaxseed and rosehip oils were characterized by good oxidative stability (low peroxide and *p*-anisidine values after 3 months of storage), in contrast to camelina oil, for which a high tocopherol content did not inhibit the oxidation (high peroxide values), indicating that the level of phytosterols was too low to hinder the progress of oxidation.

Second, a low amount of tocopherols (260–420 mg/kg), even in oils characterized by a low content of an oxidatively unstable ALA and high content of LA, does not inhibit the progress of oils’ oxidative deterioration (pumpkin and milk thistle oils) regardless of the domination of α- or γ-tocopherol and sterol content. In oils characterized by the high content of LA (at least 50% as in walnut, milk thistle, and pumpkin) and low tocopherol content, we observed the accumulation of primary and secondary products of lipid oxidation. For oxidative stabilization of oils rich in linoleic acid, the recommended molar ratio of tocopherols to LA should be higher than 1:2000.

Finally, we noted that polyphenols worked in synergism with adequate phytosterol and tocopherol concentrations to prevent the progress of lipid oxidation. However, polyphenols in LA—rich oils in molar ratio up to ca. 1:3000 (polyphenols:LA) did not prevent the formation of primary and secondary lipid oxidation products, which was observed in walnut, pumpkin, and milk thistle oils. These phenomena need some clarification in further scientific investigations.

## 5. Conclusions

The antioxidant behavior of different classes of antioxidants represents a complex phenomenon depended not only on the structure of the antioxidant but also on its concentration. The phenomena are obvious for tocopherols but also evident for other antioxidants such as phytosterols, polyphenols, and flavonoids, and all of them may synergize with each other and influence the oxidative stability of the oil. We have identified combinations of antioxidants that added to cold-pressed oils could provide potent strategies to increase the oxidative stability of cold-pressed oils.

## Figures and Tables

**Figure 1 foods-09-01630-f001:**
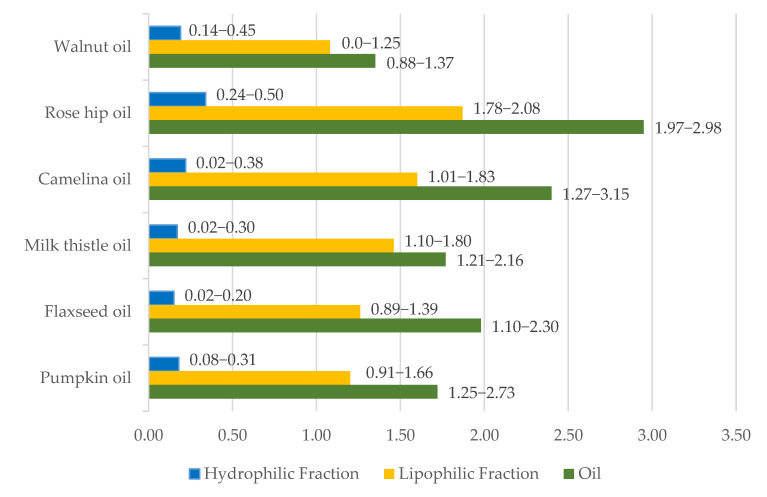
The antiradical scavenging activity of oils and their fractions—DPPH assay [TEAC, mM/kg]; median and range.

**Figure 2 foods-09-01630-f002:**
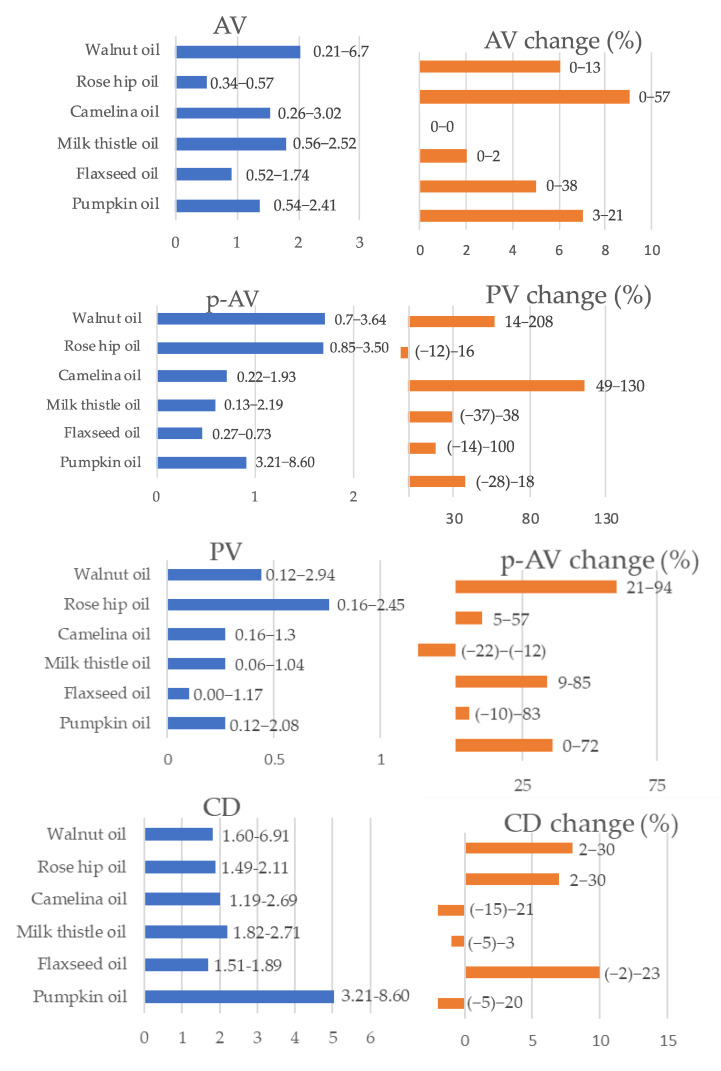
Oxidative stability parameters in fresh oils (blue bars) and % of change after 3 months of shelf life storage (orange bars); median and range. AV—acid value [mg KOH/g], PV—peroxide value [mEq O_2_/kg], *p*–AV—*p*–anisidine value, CD—conjugated diene [µmol/g], CT—conjugated triene [µmol/g].

**Table 1 foods-09-01630-t001:** Fatty acids in oils, median [%].

Fatty Acids	Walnut Oil	Rosehip Oil	Camelina Oil	Milk Thistle Oil	Flaxseed Oil	Pumpkin Oil
C16:0	7.5	4.4	5.7	8.6	6.2	12.2
	(6.6–8.2)	(4.3–4.5)	(5.6–7.0)	(7.4–9.1)	(5.1–6.7)	(10.6–13.4)
C18:0	2.6	2.1	2.6	4.9	4.4	5.8
	(2.0–3.1)	(2.1–2.1)	(2.4–3.3)	(4.6–5.4)	(4.0–5.6)	(4.7–6.2)
C18:1n-9	18.9	16.9	15.8	22.4	20.5	30.8
	(17.2–20.4)	(16.1–17.0)	(12.8–20.3)	(20.4–23.5)	(15.9–23.2)	(24.0–42.6)
C18:2n-6	58.5	43.7	17.4	57.4	15.2	49.9
	(55.3–60.7)	(43.5–43.7)	(16.1–21.6)	(55.4–61.1)	(12.5–18.2)	(39.6–54.7)
C18:3n-3	11.7	30.9	49.9	1.0	53.1	0.6
	(10.1–12.5)	(30.7–31.2)	(41.2–53.6)	(0.9–1.0)	(49.3–59.3)	(0.4–1.1)
C20:0	tr.	0.7	1.5	2.3	0.2	0.4
		(0.7–0.8)	(1.0–1.6)	(0.0–3.0)	(0.0–0.2)	(0.4–0.8)
C20:1n-9	tr.	0.4	1.6	0.6	tr.	tr.
		(0.4–0.4)	(1.1–3.7)	(0.0–0.9)			
C22:0	n.d.	n.d.	n.d.	2.1	n.d.	n.d.
				(1.9–2.3)			
C24:0	n.d.	n.d.	n.d.	0.4	n.d.	n.d.
				(0.0–0.6)			
∑ SFA	10.2	7.1	9.9	17.4	10.7	18.2
	(9.1–11.6)	(7.1–7.4)	(9.7–11.1)	(15.7–20.2)	(9.3–12.5)	(16.8–20.2)
∑ MUFA	19.1	17.3	17.1	22.7	20.5	30.8
	(17.2–20.4)	(16.5–17.4)	(15.9–22.8)	(21.2–24.2)	(16.0–23.4)	(24.0–42.6)
∑ PUFA	70.3	74.6	66.6	58.4	68.8	50.7
	(67.8–72.9)	(74.2–74.9)	(62.8–71.6)	(56.4–62.1)	(64.4–73.2)	(40.2–55.6)
n-6/n-3 ratio	5.1	1.4	0.4	57.6	0.3	80.0
	(4.4–5.7)	(1.4–1.4)	(0.3–0.5)	(56.0–62.4)	(0.2–0.3)	(48.4–110.9)

SFA—saturated fatty acid. MUFA—monounsaturated fatty acid. PUFA—polyunsaturated fatty acid. tr.—traces < 0.1%, n.d.—not detected.

**Table 2 foods-09-01630-t002:** Phytosterol content in oils, median [mg/kg of oil].

Compound	Product Ion Mas Spectra Data (*m*/*z*)					Walnut Oil	Rosehip Oil	Camelina Oil	Milk Thistle Oil	Flaxseed Oil	Pumpkin Oil
**Cholesterol**	458	443	368	353	329	64.2	n.d.	165.6	197.2	38.3	n.d.
						(41.7–83.5)		(131.4–187.5)	(166.5–302.1)	(12.6–61.2)	
**Δ^5^-Campesterol**	472	457	382	367	343	113.7	178.0	595.6	212.6	888.4	43.8
						(85.2–197.1)	(166.1–208.3)	(515.1–627.4)	(176.3–249.9)	(434.4–1266.7)	(13.5–202.4)
**Δ^5^-Stigmasterol**	484	469	394	379	355	69.3	83.1	19.4	268.2	273.8	27.5
						(5.19–94.8)	(67.1–157.4)	(16.6–26.1)	(216.8–460.3)	(104.3–413.9)	(8.1–122.3)
**Brassicasterol**	470	457	382	367	343	n.d.	n.d.	123.9	n.d.	n.d.	n.d.
								(109.4–145.9)			
**Δ^7^-Campesterol**	482	467	392	377		n.d.	n.d.	n.d.	111.7	n.d.	n.d.
									(100.3–114.7)		
**Δ^7,22,25^-Stigmastatrienol**	482	467	392	377		n.d.	n.d.	n.d.	n.d.	n.d.	1355.8
											(427.8–2121.2)
**β-Sitosterol**	486	471	396	381	357	1091.9	4314.2	1440.2	1479.7	1839.0	172.3
						(715.1–2251.4)	(3856.9–4934.1)	(1231.4–1484.9)	(1120.1–2377)	(912.6–2326.1)	(39.2–1214.7)
**α-Spinasterol**	484	469	394	379		n.d.	n.d.	n.d.	n.d.	n.d.	1412.4
											(817.6–2192.5)
**Δ^5^-Avenasterol**	484	469	394	379	355	101.1	282.2	141.7	43.6	441.5	n.d.
						(55.9–145.7)	(155.9–337)	(118.5–173.9)	(17.8–196.5)	(245.1–584.2)	
**Δ^7,25^-Stigmastadienol**	484	469	394	379		n.d.	n.d.	n.d.	n.d.	n.d.	1228.8
											(1017.4–1611.6)
**Δ^5,24^-Stigmastadienol**	484	469	394	379		n.d.	n.d.	n.d.	101.1	n.d.	n.d.
									(50.8–138.5)		
**Δ^7^-Stigmastenol**	486	471	396	381		n.d.	n.d.	n.d.	1003.1	n.d.	223.1
									605.7–1374.3		(148.6–561.5)
**Δ^7^-Avenasterol**	484	469	394	379		227.3	76.6	n.d.	151.2	n.d.	829.7
						(163.3–368.6)	(55.3–90.2)		(89.3–347.9)		(729.7–1468.9)
**Cycloartenol**	427	409	320	257	191	n.d.	288.8	24.6	n.d.	1562.3	n.d.
							(200.3–358.3)	(23.6–41.1)		(825.1–2260.6)	
Total						1421.7	5358.2	2533.1	3421.1	5171.7	5459.9
						(973.7–2880.3)	(4835.1–5837.3)	(2137.7–2755.7)	(2048.1–5501.3)	(2615.8–5979.4)	(3964.9–7977.7)

n.d.—not detected.

**Table 3 foods-09-01630-t003:** Antioxidants and other minor components of oils affecting antioxidant potential and oxidative stability of oils, median [28].

	Walnut Oil	Rosehip Oil	Camelina Oil	Milk Thistle Oil	Flaxseed Oil	Pumpkin Oil
The antioxidants content [mg/kg]						
Tocopherols						
α-Tocopherol	48.6	123.8	n.d.	204.1	63.5	51.2
	(40.3–82.0)	(81.0–164.6)		(200.2–301.6)	(22.0–225.3)	(24.8–65.3)
γ-Tocopherol	335.6	674.8	817.7	55.5	540.3	201.4
	(207.1–380.3)	(533.0–683.4)	(658.5–888.0)	(49.7–84.2)	(454.8–619.3)	(165.0–360.6)
δ-Tocopherol	45.6	252.9	126	14.6	n.d.	21.9
	(42.4–54.4)	(237.4–311.9)	(37.8–222.6)	(9.0–15.3)		(17.2–27.8)
Total tocopherols	423.1	1036.0	972.3	262.0	588.7	290.8
	(300.7–476.9)	(866.9–1159.9)	(692.5–1026.7)	(253.8–354.6)	(476.8–490.0)	(206.9–426.4)
Total phenols (CAE)	83.6	86.8	117.3	78.8	55.8	106.6
	(59.6–252.1)	(74.71–117.1)	(34.12–138.9)	(71.7–124.7)	(37.57–84.9)	(53.67–184.6)
Total flavonoids (LE)	7.6	11.6	16.7	4.53	15.4	64.2
	(1.1–13.9)	(4.8–14.9)	(11.6–46.7)	(4.23–20.9)	(8.3–20.5)	(31.6–135.5)
Squalene	58.1	203.8	22.2	65.4	n.d.	1324.3
	(9.1–251.8)	(151.5.0–214.9)	(15.2–68.9)	(41.8–185.6)		(1050.8–1787.1)
Chlorophyll and transient metal contents [mg/kg]						
Chlorophyll	0.87	0.86	3.80	1.36	0.79	6.63
	(0.00–1.64)	(0.07–0.91)	(0.03–13.44)	(0.39–2.62)	(0.32–3.37)	(1.28–13.4)
Cu	0.013	0.014	0.009	0.037	0.013	0.03
	(0.008–0.017)	(0.011–0.016)	(0.008–0.018)	(0.01–0.051)	(0.01–0.017)	(0.006–0.190)
Fe	0.23	0.08	0.14	0.19	0.16	0.40
	(0.17–0.34)	(0.04–0.25)	(0.12–0.43)	(0.16–0.44)	(0.1–1.72)	(0.19–0.84)

CAE—caffeic acid equivalent. LE—luteolin equivalent. n.d.—not detected.

**Table 4 foods-09-01630-t004:** The relation between antiradical scavenging activity values in oils and oil’s components and oxidative stability parameters assessed in the multiple regression test.

		DPPH Oil	DPPH Hydrophilic Fraction	DPPH Lipophilic Fraction
Polyphenols	b		0.480657	
	B		0.146997	
	*p*		0.002703	
Phytosterols	b			0.630663
	B			0.139384
	*p*			0.000083
Tocopherols	b	0.655721	0.443316	
	B	0.135603	0.146997	
	*p*	0.000034	0.005180	
PV after 3 months	b	−0.480145		
	B	0.671066		
	*p*	0.027654		

**Table 5 foods-09-01630-t005:** Significant effects of oil components on DPPH antioxidant activity of oil—multiple regression test results.

DPPH			Walnut Oils	Rosehip Oils	Camelina Oils	Milk Thistle Oils	Flaxseed Oils	Pumpkin Oils
Tocopherols	b	0.013669						
	B	0.003224		◉	◉	◉		◉
	*p*	0.000493						
Polyphenols	b	0.101689						
	B	0.056526	◉		◉	◉		
	*p*	*0.043608*						
Flavonoids	b	0.427534						
	B	0.136117	◉		◉			
	*p*	0.009393						
Phytosterols	b	0.005055						
	B	0.001433		◉			◉	
	*p*	0.009617						

◉—statistically significant.

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
