# Peer review of "Characteristics and Antioxidant Potential of Cold-Pressed Oils—Possible Strategies to Improve Oil Stability"

_foods, 2020, doi:10.3390/foods9111630_

Round 1

Reviewer 1 Report

The paper is good but lacks some originality. The authors have to explain why they choose these types of oils and not others?

English language requires polishing. Please ask an English native speaker to review the manuscript.

The introduction should cite some papers regarding the oxidative stability of oils, and showing the originality compared to those papers.

Below some other minor comments :

Lines 75-77: put the latin names of plants in italic

Line 81: how 12 h day light condition was applied? Please detail.

Line 102: change “ambiguous result” by “co-eluted molecules”

Line 108: add a “,” after “content”

Lines 113-114: the references are cited twice. Correct them please.

Lines 116 and 121: add “content” after “chlorophyll” and “Fe”, respectively

Lines 179-182: please detail why these compounds are prooxydant?

Line 189: this sentence should be removed

Author Response

We appreciate your time and efforts in reviewing our article. We paid all possible efforts to improve the paper according to the remarks.We have marked red the new and altered passages in the current version of the paper. To support new ideas, we added and cited five new references and removed less relevant. The manuscript has already been copy edited by a native speaker.

We do hope that the present version of our manuscript, adjusted to the Referee’s remarks, will be suitable publication in Foods.

In line with the suggestions, we amended the manuscript as listed below.

  1. The paper is good but lacks some originality. The authors have to explain why they choose these types of oils and not others?

Thank you for pointing this out, to highlight the originality of our findings we added some passages into Introduction and Discussion.

  1. lines 75-77:put the latin names of plants in italic

We put the Latin names of plants in italic according to your suggestion

  1. Line 81:how 12 h day light condition was applied? Please detail. The detailed conditions of shelf storage are now added into manuscript. “The same analyses were also performed in oils after 3 months of shelf-life storage at room temperature and a 12 h day/night light regime” into “The oxidative stability parameter analyses were also performed in oils after 3 months of shelf-life storage at room temperature at 220C +/- 10 During the shelf-life storage the oils were kept in the dark or exposed to cold fluorescent light in a cycle of 12 h light, alternating with 12 h darkness.”
  2. Line 102: change “ambiguous result” by “co-eluted molecules” According to Reviewer suggestion it has been changed from“ambiguous result” to “co-eluted molecules
  3. Line 108: add a “,” after “content”The sentences was adjusted according to the suggestion.
  4. Lines 113-114: the references are cited twice. Correct them please.We apologize for having overlooked it; the repeated reference has been deleted.
  5. Lines 116 and 121: add “content” after “chlorophyll” and “Fe”, respectively the tittle of subsections has been changed as suggested
  6. Lines 179-182: please detail why these compounds are prooxidant?We appreciate your comment and we tried to explained better the mechanism of the acceleration of lipid oxidation by transition metals and chlorophylls in a fewer passages supported by an appropriate citation:“Chlorophyll and transition metals such as iron or copper may negatively affect the oxidative stability. Metals increase the rate of oil oxidation due to the reduction of activation energy of lipids autoxidation. Chlorophylls and their degradation products may also accelerate the lipids oxidation in the presence of light and atmospheric triplet oxygen [4]. Therefore the amounts of components that may negatively affect oxidative stability were also measured. Pumpkin oils contained the most chlorophylls (6.63 mg/kg) and iron (0.4 mg/kg) among the studied oils. The highest amount of Cu was noticed in milk thistle oil (0.037 mg/kg). In other oils the median level of Cu did not exceed 0.015 mg/kg”
  7. Line 189: this sentence should be removed. We apologize for the sentence leftover from journal template, it has been removed.

Reviewer 2 Report

The manuscript is a report of fatty acid composition and profile of anti-oxidant and pro-oxidant components of 34 commercial cold-pressed oil samples in Poland. The correlations between content of bioactive compounds and oxidative stability parameters of the oil samples were also figured out and analysed.

The research was well designated and credible methods were used. Manuscript was well prepared with a clear presentation of the results and convincing discussion which emphasized the significance and novelty of the findings. The findings of this study can make practical contributions to the oil processing industry as well as further studies in the related field.

Author Response

We are encouraged and grateful for the positive feedback on our manuscript.

Reviewer 3 Report

In this work, authors analyzed 33 samples of commercial cold-pressed oils from 6 vegetal sources by means of the experimental determination of their fatty acid composition, the content of phytosterols, squalene, tocopherols, phenols, flavonoids, chlorophylls, Cu and Fe elements, antiradical scavenging activity, and oxidative stability. The objective was not only to determine the compositions but also to analyze existing interrelations between them.   Notwithstanding the merit of the work on providing vast characterization data about so many vegetal oils - which contribute to a rich portrayal and comparison basis - the manuscript raises my doubts regarding novelty and scientific soundness. Given that authors are not the first ones to study these oils, and that are using routine characterization methods, they should have evidenced better the scientific insights that justify their work to be published as a novel scientific contribution.  
However, I have no option but to suggest major revisions are introduced. I provide below suggestions to improve the work towards novelty and scientific soundness, most of not requiring extra experimental data, just search and processing of existing info.    * * *    MAIN REMARK/OPPORTUNITY FOR IMPROVEMENT:   Most the Abstract, Discussion, and Conclusion is based on the same recurrent idea: the link between DPPH and composition/performance, which was already known (and acknowledged) by the authors as prior knowledge. The main finding stated in the Abstract is later found in the document as having "previously been demonstrated " (Introduction, line 33-36; and Discussion, lines 235-236).  Hence, what the authors present as being a finding/novelty of their work is in fact a confirmation of prior knowledge, which raises concerns on the novelty requirement.

My suggestion for authors is to focus on alternative findings enabled by their extensive work, namely: 

1) geographical nuances of the oils in comparison with others in the literature (such as reported in 273-276)

2) Oil aging (progressive degradation of the properties);

3) Interrelation of the mapped oil compositions with known nutrition data for the same oils (using external data);

4) Comparison of oils composition/performance from cold-pressed method vs. other extraction methods (using external data);

5) Proposing enhanced mixture(s) of oils in order to compensate/balance their detected composition strengths and weaknesses.

I leave to authors' consideration enriching their work with some of these suggestions.

A final suggestion: authors should inform about the size of the hydrophilic (HF) and lipophilic fraction (LF) in each oil, as this parameter is itself an interesting basis for comparison of the oils.

* * * 

ADDITIONAL REMARKS:

  • Acronyms: Authors should use fewer acronyms or aid the reader by presenting their meaning more often. For instance, acronym meanings should be explicit in every table, as a final row. Secondly, in the text authors should introduce acronyms in new sections of the manuscript, otherwise, readers need to recurrently seek their meanings elsewhere in the manuscript. Please revise.

  • Line 189: " All figures and tables should be cited in the main text as Figure 1, Table 1, etc." Please confirm if this sentence is right.
    Line 284-286: " High squalene content is a corollary to technology of pumpkin oil manufacturing, where temperature and seed hulling seem to286 play a crucial role [44] " This sentence appear to convey an interesting idea but I could not understand its meaning. I suggest the authors to explain better this idea.

  • Vegetal species should be always written in italic.

  • To increase the appeal of the document, please consider presenting some of the data in figures.

Author Response

The manuscript has been revised and undergone major changes to support some ideas presented in the manuscript and justify the study as a novel scientific insight and a contribution to the field of lipid oxidation. We tried to focus according to your suggestion on two main directions proposed in the revision: mainly oil aging and a proposal of oil mixtures in order to compensate lipid deterioration. We have marked red the new and altered passages in the current version of the paper. To support new ideas, we added and cited five new references and removed less relevant. The manuscript has already been copy edited by a native speaker. We would like to thank you for your thorough review that helped us to improve the manuscript.

Additional remarks.

  1. Acronyms: Authors should use fewer acronyms or aid the reader by presenting their meaning more often. For instance, acronym meanings should be explicit in every table, as a final row. Secondly, in the text authors should introduce acronyms in new sections of the manuscript, otherwise, readers need to recurrently seek their meanings elsewhere in the manuscript. Please revise.

We amended the manuscript according to Reviewer suggestion traying to avoid unnecessary usage of acronymous.

  1. Line 189: " All figures and tables should be cited in the main text as Figure 1, Table 1, etc." Please confirm if this sentence is right.

We apologize for leaving unnecessary sentences from journal template, it has been removed.

  1. Line 284-286: " High squalene content is a corollary to technology of pumpkin oil manufacturing, where temperature and seed hulling seem to286 play a crucial role [44] " This sentence appear to convey an interesting idea but I could not understand its meaning. I suggest the authors to explain better this idea".

Thank you for pointing this out. We have changed it from: “High squalene content is a corollary to technology of pumpkin oil manufacturing, where temperature and seed hulling seem to play a crucial role [44]” to “To obtain a high squalene content in pumpkin oil the precise manufacturing conditions play a crucial role. Generally, husking and appropriately heating up seeds just before pressing results in oils with a higher squalene level” [44]”.

  1. Vegetal species should be always written in italic. We put the Latin names of plants in italicaccording to the suggestion.
  2. To increase the appeal of the document, please consider presenting some of the data in figures

We appreciate you notice. We added the figures presenting oil deterioration parameters and their changes and antioxidant potential into the manuscript.

Round 2

Reviewer 3 Report

The authors addressed the queries to a satisfactory extent.
I thank the authors for their effort to improve the manuscript.